# β-Arrestin 1 Differentially Modulates cAMP and ERK Pathways Downstream of the FSH Receptor

**DOI:** 10.3390/cimb47121051

**Published:** 2025-12-16

**Authors:** Sei Hyun Park, Munkhzaya Byambaragchaa, Ye Rin Yu, Jae Won Lee, Min-Jeong Kwak, Seung-Bin Yoon, Ji-Su Kim, Myung-Hwa Kang, Kwan-Sik Min

**Affiliations:** 1Graduate School of Animal BioScience, Hankyong National University, Anseong 17579, Republic of Korea; 2Carbon-Neutral Resources Research Center, Institute of Genetic Engineering, Hankyong National University, Anseong 17579, Republic of Korea; 3Division of Animal BioScience, School of Animal Life Convergence Sciences, Hankyong National University, Anseong 17579, Republic of Korea; 4Pimate Resources Center, Korea Research Institute of Bioscience and Biotechnology, Jeongup 56216, Republic of Korea; 5Department of Food Science and Nutrition, Hoseo University, Asan 31499, Republic of Korea

**Keywords:** β-arrestin 1, cAMP, ERK1/2, FSHR

## Abstract

This study compared the gonadotropin gene sequences (LH and FSH subunits) of Cynomolgus and Rhesus monkeys and produced recombinant single-chain LHβ/α and FSHβ/α proteins. The α- and FSHβ-subunit sequences were identical between species, while LHβ showed only minor synonymous differences. The recombinant hormones were successfully expressed and shown to be mainly N-glycosylated. Recombinant monkey FSHβ/α activated cAMP signaling in human FSH receptor-expressing cells, confirming its biological activity. β-arrestin 1 was found to have dual roles: its absence increased cAMP signaling (negative regulation), but it was required for ERK1/2 activation. ERK activation depended mainly on the cAMP/PKA pathway. Human and rat FSH receptors displayed different ERK activation timing, indicating species-specific signaling behavior. Overall, the study establishes a reliable system for producing functional recombinant monkey gonadotropins and clarifies how β-arrestin 1 differentially regulates FSH receptor signaling.

## 1. Introduction

Follicle-stimulating hormone (FSH) is a member of the glycoprotein hormone family, which also includes luteinizing hormone (LH), thyrotropin (TSH), and chorionic gonadotropin (CG). These hormones are heterodimeric proteins composed of a common α-subunit and a hormone-specific β-subunit that confers biological specificity [1]. Among them, FSH, LH, and TSH are synthesized and secreted by the anterior pituitary gland, whereas CG is produced by the placenta. FSH and LH secretion exhibit a coordinated pattern, particularly during the preovulatory surge. Collectively, glycoprotein hormones play essential roles in regulating mammalian reproductive and endocrine physiology.

Several studies have examined recombinant human FSH (rec-hFSH) produced in mammalian expression systems and characterized its biological activity, particularly regarding the removal or modification of carbohydrate residues [2,3,4]. The FSH receptor (FSHR) is primarily expressed in ovarian granulosa cells and testicular Sertoli cells, where it mediates the effects of FSH on gametogenesis and steroidogenesis [5]. Upon ligand binding, FSHR undergoes conformational changes that activate heterotrimeric G proteins, predominantly Gαs, leading to stimulation of adenylyl cyclase and subsequent production of cyclic adenosine monophosphate (cAMP) [6]. This classical G-protein-mediated signaling pathway triggers downstream events, including protein kinase A (PKA) activation and transcriptional regulation, which are required for follicular development and spermatogenesis. Recently, cryo-electron microscopy structures of the FSH–FSHR–Gαs and CG–LH/CGR–Gαs complexes have been elucidated, providing key insights into hormone specificity and the molecular mechanisms of receptor activation [7,8].

G-protein-coupled receptors (GPCRs) constitute the largest family of transmembrane cell-surface receptors and regulate diverse intracellular signaling pathways in response to a wide range of extracellular stimuli [9,10]. Gonadotropin receptors, including FSHR, belong to this superfamily and are characterized by seven transmembrane-spanning domains and extensive extracellular and intracellular regions that are critical for ligand binding and receptor signaling [11]. Besides the classical G-protein-dependent pathway, FSHR also activates β-arrestin-mediated signaling, which contributes to extracellular signal-regulated kinase 1/2 (ERK1/2) activation and receptor internalization, providing spatial and temporal regulation of receptor responses [9,12]. This dual signaling capacity of FSHR enables precise modulation of reproductive physiology, underscoring the importance of understanding both G-protein- and β-arrestin-dependent pathways in the context of gonadotropin action.

Following receptor activation, β-arrestins are recruited to FSHR after phosphorylation of its intracellular regions by GPCR kinases (GRKs) [13,14]. Inhibition of GRK2 markedly enhances FSHR-mediated ERK1/2 phosphorylation, whereas GRK5 and GRK6 are required specifically for β-arrestin-dependent ERK activation [15,16]. Thus, β-arrestins bind to the phosphorylated C-terminal tail of FSHR, promoting receptor internalization and initiating ERK signaling, independent of G-protein activation [17]. Although Gαs is not required for β-arrestin coupling, it influences GRK recruitment selectivity and predominantly contributes to downstream gene expression regulation in other GPCR systems, such as the β2-adrenergic receptor [18]. Therefore, a detailed understanding of β-arrestin-mediated signaling is essential for elucidating the full spectrum of FSHR downstream regulatory mechanisms.

We previously reported that the biological activities of glycoprotein hormones, including equine CG (eCG), equine FSH (eFSH), and eel FSH, are critically influenced by the composition and structure of their attached oligosaccharide chains, which play pivotal roles in receptor-mediated signal transduction [19,20,21]. Although the cloning of monkey FSH cDNA has been described in earlier studies [22], we re-examined and verified the correct cDNA sequences of these gonadotropins to ensure accuracy in subsequent structural and functional analyses. Due to the recent increase in new drug development, the demand for monkeys has risen sharply, leading to a growing need for monkey FSH for controlled ovarian stimulation. Therefore, it is necessary to develop a large-scale production system for monkey FSH to meet this demand.

In this study, we demonstrated that recombinant monkey FSH (rec-monkey FSH) elicited a full cAMP response in cells expressing the human FSHR (hFSHR), and notably, cAMP levels were higher in β-arrestin 1 knockout (KO) cells than in wild-type cells. In contrast, ERK1/2 phosphorylation (pERK1/2) was delayed in cells expressing rat FSHR (rFSHR) compared to those expressing hFSHR, indicating species-specific differences in receptor signaling kinetics. Furthermore, pERK1/2 activation was abolished in β-arrestin 1 KO cells, demonstrating that β-arrestin 1 is essential for sustaining ERK1/2 signaling downstream of the FSHR.

## 2. Materials and Methods

### 2.1. Materials

The TRIzol reagent for total RNA extraction and the 1st strand cDNA synthesis kit were purchased from Gibco BRL (Grand Island, NY, USA). Oligonucleotides were synthesized, and cloned DNAs were sequenced by Genotech (Daejeon, Republic of Korea). The pGEM-T Easy Cloning Vector and Gel Purification kit was from Promega (Madison, WI, USA). Polymerase chain reaction (PCR) reagents, Ex-tag, DNA ligation reagents, and restriction enzymes were purchased from Takara Bio (Shiga, Japan). CHO-K1 and HEK293 cells were obtained from the Korean Cell Line Bank (Seoul, Republic of Korea). Ham’s F-12 medium, Opti-MEM I, Dulbecco’s modified Eagle’s medium, HEPES, gentamycin, Fetal Bovine Serum (FBS) and serum-free CHO-S-SFM II were purchased from Gibco (Grand Island, NY, USA). The mammalian expression vectors pcDNA3.1 and pcDNA3.1(−)/Myc-His, along with CHO suspension (CHO-S) cells, FreeStyle MAX transfection reagent, Lipofectamine 2000, His-Tag Monoclonal Antibody (MA1-21315), and FreeStyle CHO expression medium were purchased from Invitrogen Corporation (Carlsbad, CA, USA). PNGase F (P0704), O-Glycosidase (P0733), and α2-3,6,8,9 Neuraminidase A (P0722) were purchased from New England Biolabs (Ipswich, MA, USA). The cAMP Homogeneous Time-Resolved Fluorescence (HTRF) assay kit, developed for the sensitive and quantitative detection of intracellular cAMP levels, was obtained from Cisbio (Codolet, France). Anti-phospho-p44/42 MAPK, total ERK1/2 antibodies, and β-arrestin 1 (D7Z3W) XP monoclonal antibody were purchased from Cell Signaling Technology (Beverly, MA, USA). HisPur^TM^ Ni-NTA Spin Columns, Strep-tag polyclonal antibody (PA5-114454) and SuperSignal^TM^ enzyme-linked immunosorbent assay (ELISA) Femto Maximum Sensitivity Substrate were purchased from Thermo Fisher Scientific Inc. (Rockford, IL, USA). The beta-actin polyclonal antibody (ab8227) was purchased from Abcam (Cambridge, UK). The Strep-Tactin resin column was purchased from IBA Lifesciences GmbH (Gottingen, Germany).

The mammalian expression vector pCORON1000 SP VSV-G, which contains a VSV-G signal peptide to enhance protein expression and secretion, was obtained from Amersham Biosciences (Piscataway, NJ, USA). β-Arrestin-1 CRISPR/Cas9 KO Plasmid (h), β-Arrestin-1 homology-directed repair (HDR) Plasmid, and ERK1/2 (C-9) antibody were purchased from Santa Cruz Biotechnology, Inc. (Dallas, TX, USA). H-89 dihydrochloride (H-89) and GF 109203X (GFX) were purchased from TOCRIS Bioscience (Bristol, UK). The QIAprep-Spin plasmid kit was purchased from Qiagen Inc. (Hilden, Germany), and disposable flasks were purchased from Corning Inc. (Corning, NY, USA). CentriPlus centrifugal filters were purchased from Amicon Bio Separation (Billerica, MA, USA). All other reagents were purchased from Sigma-Aldrich (St. Louis, MO, USA).

### 2.2. Total RNA Isolation and cDNA Synthesis

Pituitary tissues were collected from *Cynomolgus monkey* (*Macaca fascicularis*) and *Rhesus monkey* (*Macaca mulatta*) species, which were provided by the Primate Resources Center at the Korea Research Institute of Bioscience and Biotechnology (KRIBB). The tissues were quickly frozen in liquid nitrogen and stored at −80 °C. To extract total RNA, the tissues were ground using TRIzol reagent. RNA was dissolved in RNase-free water and stored at −80 °C. The amount and quality of RNA were checked using a NanoDrop and by running a gel. Five micrograms of RNA was used to prepare cDNA. First, RNA was mixed with Oligo (dT) primers and reaction buffer and heated at 42 °C for 2 min. Next, 1 µL of SuperScript II enzyme was added, and the mixture was kept at 42 °C for 50 min for reverse transcription. Finally, RNase H was added and incubated at 37 °C for 20 min to remove RNA. All experimental procedures involving the use of monkey tissues were reviewed and approved by the Institutional Animal Care and Use Committee (IACUC) of KRIBB, in accordance with established ethical guidelines (approval no. KRIBB-AEC-23011). The study was conducted in compliance with all institutional and national regulations regarding the care and use of laboratory animals.

### 2.3. PCR Amplification of Monkey α-, LHβ- and FSHβ-Subunit cDNAs

The cDNA was subsequently used as a template for PCR amplification of the α-, LHβ-, and FSHβ-subunit cDNAs. Gene-specific primers for each of the three subunits were designed based on previously published sequences [22]. Previous studies have indicated that the glycoprotein hormone sequences differ between cynomolgus and rhesus monkeys. To confirm these observations, we conducted an additional comparative sequence analysis of the two species. To facilitate subsequent cloning steps, restriction enzyme recognition sites for EcoRI and SalI or XhoI were incorporated into the 5′ and 3′ ends of the primers. All PCR primers used in this experiment are provided in the attachment (Appendix A). These sites allowed directional cloning into expression vectors in subsequent experiments. PCR amplification was conducted under optimized conditions to ensure the specific amplification of the target cDNA fragments. Each reaction produced a single band of the expected size on agarose gel electrophoresis, confirming the specificity of the primers and quality of the cDNA. The PCR products were purified using a gel extraction kit to remove residual primers and enzymes and were subsequently subcloned into the pGEM-T Easy vector for sequencing. For each gene, approximately 10 independent clones from both Cynomolgus and Rhesus monkey samples were selected and subjected to sequencing. The obtained sequences were aligned and analyzed to identify the nucleotide differences between the two monkey species. Sequence alignments and comparative analyses were performed using bioinformatics tools, such as the Multalin multiple sequence alignment tool.

### 2.4. Construction of Single-Chain Monkey LHβ/α and FSHβ/α

To construct single chains of LHβ/α and FSHβ/α, we used an overlap extension PCR method to attach the common α-subunit to the C-terminal region of the LHβ- and FSHβ-subunits, as described previously [23]. Briefly, two sets of PCR primers were synthesized. In Step I, the first fragment was amplified using forward and reverse primers that included the C-terminal region of the LH β-subunit and the mature protein region of the α-subunit. The second fragment was amplified using a forward primer that included the C-terminal region of the LH β-subunit and mature protein region of the α-subunit and reverse primers including the stop codon of the α-subunit. In Step II, the two fragments obtained in Step I were fused using overlap extension PCR. Single-chain FSHβ/α was constructed using the same strategy as for LHβ/α. The resulting PCR products were ligated into the pGEM-T Easy vector, and the sequences of the single-chain monkey LHβ/α and FSHβ/α constructs were confirmed via DNA sequencing.

### 2.5. Construction of Mammalian Expressing Vectors

First, full-length fragments were cleaved using EcoRI and SalI restriction enzymes and ligated into the pcDNA3 expression vector. The constructs were inserted into the pcDNA3.1(−)/Myc-His vector using EcoRI and XbaI (or KpnI) restriction sites to incorporate Myc and His epitope tags without a stop codon. Subsequently, to efficiently detect and purify rec-monkey FSH, we introduced four amino acids (GGSA) and Strep- or His-tag sequences at the C-terminal region of the α-subunit (Figure 1). The modified genes were then subcloned into the pcDNA3.1 mammalian expression vector. The following expression constructs were generated: pcDNA3.1-M-LH β/α and pcDNA3.1-M-FSH β/α (for untagged versions), pcDNA3.1(-)/Myc-His-M-LH β/α and pcDNA3.1(-)/Myc-His-M-FSH β/α (for Myc-His-tagged versions), and pcDNA3.1-M-FSH β/α (for expression in CHO-S cells; GGSA-His and Strep-tag versions).

### 2.6. Production of Rec-Monkey LHβ/α and FSH β/α into CHO-K1 and CHO Suspension Cells

Plasmids cloned into the pcDNA3.1(-)/Myc-His expression vectors were transfected into CHO-K1 cells using liposome-mediated transfection, as described previously [23]. Cells were seeded in 6-well plates and grown to 80–90% confluence. After washing with Opti-MEM to remove residual serum and antibiotics, DNA–Lipofectamine complexes were added to Opti-MEM. After 5 h of incubation, the transfection medium was replaced with fresh growth medium containing 20% FBS to promote cell recovery. The next day, the cells were washed twice with Opti-MEM I and cultured in serum-free CHO-S-SFM II medium. Supernatants were collected 72 h post-transfection to assess the expression of rec-monkey LHβ/α and FSHβ/α proteins. pcDNA3-TOPO mammalian vectors were transfected into CHO-S suspension cells using the FreeStyle MAX reagent, as previously described [23]. Briefly, CHO-S cells were cultured in 30 mL of CHO expression medium under standard conditions (37 °C, 8% CO_2_, shaking at 120 rpm) for 3 days.

Following this initial culture, the cells were subcultured at a density of 5–6 × 10^5^ cells/mL into 50–125 mL of fresh CHO expression medium in a disposable spinner flask and maintained under continuous agitation (120 rpm) to support suspension growth. On the day of transfection, the cell density reached approximately 1.2–1.5 × 10^6^ cells/mL. For transfection complex formation, 160 µg of plasmid DNA and 160 µL of FreeStyle MAX reagent were diluted in 1.2 mL of OptiPRO serum-free medium, gently mixed, and incubated at room temperature for 5 min. The resulting transfection mixture (total volume: 2.4 mL) was added dropwise to the spinner culture to ensure even distribution. Culture supernatants were harvested on day 9 post-transfection and clarified by centrifugation at 500× *g* for 5 min. The collected media were concentrated using centrifugal filter units according to the manufacturer’s instructions. The expression of recombinant proteins was confirmed using Western blot analysis.

### 2.7. Purification of Strep- and His-Tagged Rec-Monkey FSH Protein

Protein purification was conducted using both gravity-flow and spin-column-based methods. For Strep-Tactin affinity purification, the column was first equilibrated with 5 mL of Buffer W to prepare the resin for binding. After equilibration, 50 mL of the clarified protein sample was applied to the column via gravity flow. The column was then washed five times with 5 mL of Buffer W to remove non-specifically bound proteins and other contaminants. Elution was performed in six sequential steps using 2.5 mL aliquots of Buffer E, which contained 2.5 mM desthiobiotin, a competitive ligand that specifically displaces Strep-tagged proteins from the resin. This stepwise elution strategy ensured the efficient recovery of the target protein with minimal background contamination.

HisPur™ Ni-NTA spin columns were used to purify polyhistidine-tagged proteins via immobilized nickel ions on the resin. The columns were first equilibrated with two resin-bed volumes of equilibration buffer comprising 20 mM sodium phosphate, 30 mM sodium chloride, and 10 mM imidazole (pH 7.4) to condition the resin for optimal binding. Following equilibration, excess buffer was removed by centrifugation at 700× *g* for 2 min. The clarified protein-containing supernatant was gently applied to the column. To enhance the binding between the His-tagged protein and the Ni^2+^-charged resin, the column was incubated on an orbital shaker for 30 min at either room temperature (RT) or 4 °C, depending on the experimental requirements. After incubation, non-specifically bound proteins were removed by washing the column twice with a wash buffer composed of phosphate-buffered saline (PBS) supplemented with 25 mM imidazole (pH 7.4). The low imidazole concentration selectively eluted weakly bound contaminants while retaining the His-tagged target protein on the resin. Finally, the recombinant monkey FSHβ/α protein was eluted in three successive steps, each using 1 mL of elution buffer composed of PBS supplemented with 250 mM imidazole (pH 7.4). The high concentration of imidazole competitively displaced the His-tagged proteins from the Ni-NTA resin, enabling the efficient recovery of the target protein in the eluate.

### 2.8. Western Blotting Analysis of Rec-Monkey LHβ/α and FSHβ/α

Recombinant proteins expressed in CHO-K1 and CHO-S cells were analyzed by concentrating culture supernatants or purified eluates, respectively, followed by SDS-PAGE, membrane transfer, and immunodetection using monoclonal antibodies against Myc, His, or Strep tags to confirm protein expression. After incubation with primary antibodies, membranes were further incubated for 1 h at RT with horseradish peroxidase (HRP)-conjugated secondary antibodies to enable signal amplification. Protein bands were visualized using the SuperSignal™ West Femto Maximum Sensitivity Substrate (Thermo Fisher Scientific) according to the manufacturer’s instructions, and chemiluminescent signals were captured on X-ray film for sensitive detection of recombinant monkey FSH β/α protein.

### 2.9. Enzymatic Deglycosylation of Rec-Monkey FSHβ/α

Deglycosylation of recombinant proteins was performed to characterize the presence and nature of N- and O-linked glycans, as well as terminal sialic acid residues. To remove N-linked oligosaccharides, 20 µg of recombinant protein was denatured by heating at 100 °C for 10 min in the presence of 1 µL of 10× Glycoprotein Denaturing Buffer (New England Biolabs; Ipswich, MA, USA). The denatured protein was subsequently incubated at 37 °C for 1 h in a final reaction volume of 20 µL containing 1 µL of PNGase F (2.5 U/mL), 2 µL of 10× GlycoBuffer, and 2 µL of 10% NP-40. For O-linked deglycosylation, an equivalent amount of protein (20 µg) was treated with O-Glycosidase (Endo-α-N-acetylgalactos). The protein sample was first denatured by mixing with 1 µL of 10× Glycoprotein Denaturing Buffer and heating at 100 °C for 10 min. After denaturation, the reaction volume was adjusted to 20 µL by adding 2 µL of 10× GlycoBuffer 2, 2 µL of 10% NP-40, and 2 µL of O-Glycosidase. The mixture was then incubated overnight at 37 °C to allow complete removal of O-linked glycans. In a separate experiment, terminal sialic acid residues were removed using neuraminidase (also known as acetylneuraminyl hydrolase or sialidase). For this reaction, 5 µg of protein was mixed with 2 µL of 10× GlycoBuffer in a final volume of 20 µL, followed by the addition of 5 µL of α2-3,6,8,9 Neuraminidase. The mixture was incubated at 37 °C for 1 h to induce desialylation. Following enzymatic deglycosylation treatments, the samples were analyzed by SDS-PAGE to detect mobility shifts corresponding to glycan removal. Proteins were subsequently transferred to nitrocellulose membranes and subjected to Western blot analysis using antibodies specific to the recombinant protein to confirm the efficiency of deglycosylation.

### 2.10. Transient Transfection with hLH/CGR, hFSHR, rLH/CGR, and rFSHR cDNA Plasmids

Native signal sequence regions were deleted from the cDNA constructs of hLH/CGR, hFSHR, rLH/CGR, and rFSHR. The resulting fragments encoding the mature receptor sequences were subcloned downstream of the vesicular stomatitis virus glycoprotein (VSV-G) signal sequence in a mammalian expression vector to ensure efficient secretion and membrane targeting. The correct orientation and insertion of the fragments were verified using restriction enzyme digestion. Plasmids were transfected into CHO-K1 and HEK 293 cells using liposome-mediated transfection, as described previously [23]. Cells were seeded in 6-well plates, and DNA–liposome complexes were carefully added dropwise to each well containing cells in Opti-MEM I medium. After washing the cells twice with Opti-MEM I, the culture medium was replaced with a fresh medium the following day. Cells were harvested for cAMP and pERK1/2 assays 48–72 h post-transfection.

### 2.11. Measurement of cAMP Accumulation in CHO-K1 Cells Expressing Gonadotropin Receptors

Intracellular cAMP levels were measured to assess the activation of human and rat gonadotropin receptors by the recombinant monkey ligands. HEK293 cells expressing hLH/CGR, hFSHR, rLH/CGR, or rFSHR were analyzed using the cAMP Dynamics 2 kit (Cisbio Bioassays, Codolet, France), according to the manufacturer’s instructions. Cells were preincubated with 0.5 mM IBMX to prevent cAMP degradation and then stimulated with recombinant ligands at RT for 30 min using increasing volumes of rec-monkey FSH (0.01, 0.03, 0.09, 0.1, 0.3, 0.9, 1, 3, 9, 12, and 15 µL). Subsequently, 5 µL of cAMP-d2 reagent was added to all wells except the negative controls, followed by 5 µL of cAMP Eu-cryptate antibody. The plate was then incubated at RT in the dark for 1 h. cAMP levels were determined by measuring fluorescence at 665 and 620 nm using an HTRF microplate reader (TriStar^2^ S LB 942, BERTHOLD Technologies GmbH & Co, Bad Wildbad, Germany). The signal ratio (665/620 nm) was converted to Delta F% relative to the mock-transfected controls. Standard curves were generated to calculate the cAMP concentrations using GraphPad Prism software (version 6.0).

### 2.12. Construction of Beta-Arrestin 1 Knockout Cell Lines

To generate a stable β-arrestin 1 KO cell line via HDR, HEK 293 cells were co-transfected with a CRISPR/Cas9 KO plasmid and β-arrestin 1 HDR template plasmid (Santa Cruz Biotechnology, Dallas, TX, USA). The CRISPR/Cas9 KO plasmid was designed to introduce double-strand breaks within the *ARRB1* gene locus (encoding β-arrestin 1), whereas the HDR plasmid provided a donor template for precise genome editing through homologous recombination. HEK 293 cells were seeded into 6-well plates at a density that ensured 70–80% confluency on the following day, which was optimal for transfection. For each well, 1 µg of total plasmid DNA (an equal mixture of CRISPR/Cas9 KO and HDR plasmids) was transfected using Lipofectamine according to the manufacturer’s protocol. Each component (KO plasmid, HDR plasmid, and transfection reagent) was diluted in Opti-MEM I reduced-serum medium, gently mixed, and incubated for 15 min at RT to allow DNA–lipid complex formation. Before transfection, HEK 293 cells were washed twice with Opti-MEM I to remove the serum-containing medium that could interfere with complex uptake. DNA–lipid complexes were then added dropwise to each well and incubated with the cells for 5 h. After incubation, a complete growth medium containing 20% FBS was added to support recovery. The next day, the medium was replaced with fresh complete medium to remove the residual transfection reagent.

At 72 h post-transfection, transfected cells were selected by replacing the culture medium with 1 µg/mL puromycin, which was maintained for approximately 10 days, and non-transfected cells were eliminated. Surviving cells were trypsinized and reseeded at clonal densities of 100 and 500 cells per dish to promote the formation of single-cell-derived colonies. After incubation for approximately 2 weeks, distinct colonies were visible and manually transferred into 96-well plates for clonal expansion. To confirm the successful knockout of β-arrestin 1, protein expression in expanded clones was evaluated by Western blotting using a mouse monoclonal anti-β-arrestin 1 antibody. Clones lacking detectable β-arrestin 1 protein expression were confirmed as β-arrestin 1 KO cell lines and were expanded for subsequent functional analyses.

### 2.13. Phospho-ERK1/2 Analysis

Forty-eight hours after transfection, HEK293 cells were serum-starved for 6 h and then stimulated with recombinant monkey FSH or LH for the indicated time periods. After stimulation, the cells were washed with ice-cold PBS and lysed in RIPA buffer supplemented with protease and phosphatase inhibitors. To investigate the involvement of the PKA and PKC signaling pathways, receptor-expressing cells were pretreated with the PKA inhibitor H89 (20 µM) for 15 min or the PKC inhibitor GF109203X (GFX; 2.5 µM) for 15 min before ligand stimulation. Cell lysates were gently agitated for 30 min at 4 °C and centrifuged at 16,000× *g* for 20 min to remove insoluble debris. The resulting supernatants were collected and stored at −80 °C until further analysis. Equal amounts of protein (5–20 µg) were separated using SDS-PAGE and transferred to nitrocellulose membranes for Western blotting. Membranes were probed with anti-phospho-p44/42 MAPK (ERK1/2) antibody (1:2000) and anti-MAPK1/2 antibody (1:3000) to detect phosphorylated and total ERK1/2, respectively. After incubation with HRP-conjugated anti-mouse secondary antibody for 1 h at RT, blots were developed using the SuperSignal™ West Femto Maximum Sensitivity Substrate (Thermo Fisher Scientific). Chemiluminescent signals were visualized by exposure to X-ray film, and band intensities were quantified using the Image Lab software version 6.0 (Bio-Rad, Hercules, CA, USA).

### 2.14. Data Analysis

Sequence alignment analysis was performed using the Multalin multiple sequence alignment tool to compare DNA sequences. Dose–response curves were generated from experiments performed in duplicate, and data analysis was conducted using GraphPad Prism (version 6.0; GraphPad Software, San Diego, CA, USA). cAMP responses, half-maximal effective concentration (EC_50_) values, and stimulation curves were calculated accordingly. For each experiment, the curves were normalized to the background signals obtained from mock-transfected control cells. The final curves represent the average data from three independent experiments, ensuring their reproducibility and statistical robustness. Quantitative analysis of pERK1/2 expression levels was performed by densitometric measurements of immunoblot bands. The resulting values were plotted using GraFit software (version 5.0; Erithacus Software, Horley, Surrey, UK) to generate graphical representations of the signaling responses.

Statistical analyses were performed using two-way analysis of variance (ANOVA), followed by Tukey’s post hoc test for multiple comparisons, as indicated in the figure legends. All analyses were conducted using GraphPad Prism (version 6.0). Statistical significance was determined at the following thresholds: * *p <* 0.05 and ** *p <* 0.01, indicating significant differences between groups.

## 3. Results

### 3.1. Comparative Sequence Analysis of Gonadotropin Subunit cDNAs Between Cynomolgus and Rhesus Monkeys

PCR amplification yielded products of approximately 375 bp for the common α-subunit, 437 bp for the LHβ-subunit, and 404 bp for the FSHβ-subunit, each containing a Kozak consensus sequence to enhance translational efficiency in subsequent expression experiments (Figure 2). Full-length cDNA sequences of the common α-, LHβ-, and FSHβ-subunits were successfully amplified from Cynomolgus monkeys (*Macaca fascicularis*) and Rhesus monkeys (*Macaca mulatta*). To confirm sequence accuracy and assess allelic variation, approximately 10 independent clones per subunit were sequenced using Sanger sequencing.

Previous reports have described interspecies variability in the common α-subunit at amino acid positions 44, 77, and 85 among primates. However, in our analysis, no sequence differences were detected between cynomolgus and Rhesus monkeys. The nucleotide and corresponding amino acid sequences were conserved in all the analyzed clones. Specifically, position 44 was consistently Val (GTA) rather than Leu (TTA), position 77 was Ser (AGT) instead of Asn (AAT), and position 85 was Gln (CAG), confirming the preserved sequence across both species.

For the LHβ-subunit, we focused on amino acid positions 32 and 60, which have been reported as polymorphic sites in nonhuman primates [22]. Sequencing of 10 independent clones revealed that position 60 was encoded as Arg (CGC) rather than His (CAC). Additional nucleotide differences were observed near codons corresponding to amino acids 26, 80, and 100; however, these variations were silent and did not result in changes at the amino acid level.

In the case of the FSHβ-subunit, all sequenced clones from both monkey species exhibited nucleotide and amino acid sequences identical to those previously reported. Comparative alignment with the human FSHβ-subunit revealed only three amino acid substitutions, indicating that the FSHβ-subunit is evolutionarily more conserved than the common α or LHβ subunits.

### 3.2. Production of Recombinant LHβ/α and FSHβ/α Proteins in CHO-K1 cells

To produce recombinant proteins, plasmids encoding LHβ/α and FSHβ/α were cloned into the mammalian expression vector pcDNA3.1(-)/Myc-His and transiently transfected into CHO-K1 cells. Each construct contained the corresponding β- and α-subunit genes fused to C-terminal Myc and His epitope tags. Protein expression was assessed using Western blotting following SDS-PAGE separation (Figure 3A).

Recombinant LHβ/α (rec-LHβ/α) was successfully detected using both anti-Myc and anti-His monoclonal antibodies, confirming the expression and proper assembly of the tagged heterodimer. The apparent molecular mass of rec-LHβ/α was approximately 37–39 kDa, consistent with the predicted size, including post-translational modifications such as N-linked glycosylation. Treatment with peptide-N-glycosidase F (PNGase F) reduced the molecular mass to approximately 33–35 kDa (Figure 3B), confirming the presence of N-linked oligosaccharide chains that contributed to the higher molecular mass observed in untreated samples. These findings indicate that rec-LHβ/α expressed in CHO cells undergoes appropriate glycosylation, consistent with the characteristics of biologically active gonadotropins.

rec-FSHβ/α was weakly detected with the anti-Myc antibody, and detection with the anti-His antibody was minimal or absent (Figure 3C). This likely reflects either lower expression efficiency, reduced accessibility or stability of the C-terminal tags, or possible differences in the folding or assembly of the β/α heterodimer. At this stage, the limited detection makes it difficult to draw definitive conclusions regarding the expression level or post-translational maturation of rec-FSHβ/α.

### 3.3. Optimization of Recombinant FSHβ/α Production and Purification Using Dual Epitope Tagging in CHO-K1 and CHO Suspension Cells

To improve both the expression yield and purification efficiency of rec-FSHβ/α, the C-terminal region of the common α-subunit was engineered to contain dual-affinity tags (Strep-tag and His-tag), as described in the Materials and Methods. This dual-tagging strategy enables flexible detection and purification using two independent affinity chromatography systems. The pcDNA3.1 expression constructs were transiently transfected into CHO-K1 and CHO-S cells, and rec-FSHβ/α expression was evaluated.

In CHO-K1 cells, Western blot analysis confirmed the successful expression of rec-FSHβ/α, which was detected using both anti-Strep-tag and anti-His-tag antibodies. The apparent molecular mass ranged from approximately 38 to 41 kDa, consistent with the predicted size of the β/α heterodimer, including post-translational glycosylation (Figure 4A). To support large-scale production, the expression was conducted in suspension-adapted CHO-S cells, allowing for the collection and purification directly from the culture supernatants.

Secreted rec-FSHβ/α was purified using Strep-Tactin and Ni-NTA affinity columns. Western blot analysis of eluted fractions revealed distinct purification profiles for each tag. During Strep-Tactin purification, the target protein was most prominently detected in the fourth elution fraction, with a weaker signal observed in the third fraction, suggesting a more gradual elution profile under the applied conditions (Figure 4B). In contrast, Ni-NTA purification yielded strong signals in the first and second elution fractions, indicating efficient and concentrated recovery of His-tagged rec-FSHβ/α during the early stages of elution.

### 3.4. Enzymatic Deglycosylation Reveals N-Linked Glycosylation as the Predominant Modification in Recombinant Monkey FSHβ/α

To characterize the glycosylation profile of purified rec-FSHβ/α, enzymatic deglycosylation assays were performed using glycosidases that target N- and O-linked glycans. Treatment with PNGase F, which selectively cleaves N-linked oligosaccharides by hydrolyzing the amide bond between the asparagine residue and the innermost N-acetylglucosamine of the glycan core, resulted in a marked reduction in the apparent molecular weight of rec-FSHβ/α from approximately 38–41 kDa to 25–26 kDa. This substantial decrease confirmed the presence of extensive N-linked glycosylation on the recombinant protein, consistent with the known glycosylation sites on both the FSHβ- and common α-subunits (Figure 5A).

Treatment with O-Glycosidase (Endo-α-N-acetylgalactosaminidase), which removes core 1 and core 3 O-linked glycans, did not produce a detectable shift in molecular weight. Similarly, neuraminidase (sialidase), which cleaves terminal sialic acid residues (N-acetylneuraminic acid), did not result in an appreciable reduction in size compared to the untreated controls (Figure 5B). These findings indicate that rec-FSHβ/α expressed in CHO-S cells contains minimal O-linked glycosylation and that terminal sialylation does not significantly contribute to its electrophoretic mobility.

Taken together, the pronounced molecular weight reduction following PNGase F digestion, combined with the lack of change after O-Glycosidase and Neuraminidase treatment, demonstrates that rec-FSHβ/α glycosylation in this expression system is predominantly, if not exclusively, N-linked. This profile is consistent with the established glycosylation characteristics of native monkey FSH, in which both subunits contain N-linked but not O-linked glycosylation sites.

### 3.5. Quantification of Recombinant Monkey LHβ/α and FSHβ/α Using Commercial ELISA Kits

To determine the concentrations of rec-monkey LHβ/α and FSHβ/α, commercially available ELISA kits designed for the detection of endogenous monkey gonadotropins were used. ELISA kits from MyBioSource and LSBio, marketed for quantifying native LH and FSH in biological samples such as serum and plasma, were also tested. Despite strict adherence to the manufacturers’ protocols and testing across multiple concentrations of purified recombinant proteins, neither rec-monkey LHβ/α nor rec-monkey FSHβ/α yielded detectable signals. Multiple replicates and independent experiments confirmed that the lack of detection was not due to technical errors.

These findings indicate that commercial ELISA kits are not compatible with the recombinant single-chain fusion forms of monkey gonadotropins used in this study. Because the β- and α-subunits are covalently linked via a flexible peptide linker, the conformational epitopes recognized by the kit antibodies—optimized for detecting the native, non-fusion heterodimer—are likely to be inaccessible or structurally preserved. Consequently, absolute quantification of rec-monkey LHβ/α and FSHβ/α could not be achieved using these assays.

This limitation should be considered when interpreting subsequent analyses of biological activity and highlights the need for either custom antibody development or mass spectrometry-based quantification approaches in future studies.

### 3.6. cAMP Analysis

The in vitro activity of rec-monkey FSHβ/α was evaluated using HEK 293 cells expressing hFSHR. As shown in Figure 6, rec-monkey FSHβ/α stimulated a dose-dependent increase in cAMP production. The half-maximal effective concentration (EC_50_) and maximal response (Rmax) in cells expressing hFSHR were 4.9 µL/mL and 66.7 ± 2.4 nM per 10^4^ cells, respectively (Table 1). In β-arrestin 1 knockout (KO) cell lines, the EC_50_ values for KO #1 and KO #2 were 0.5 and 1.2 µL/mL, respectively. The Rmax values in these KO lines were also elevated, reaching 101.5 ± 1.7 and 104.2 ± 1.9 nM per 10^4^ cells. The dose–response curves for the β-arrestin 1 KO cell lines were slightly shifted to the left, with EC_50_ values approximately 4.1- to 9.8-fold lower than those of the wild-type (WT) cell line. The Rmax values in the KO lines were also approximately 1.5 times higher than those in the WT cells. These findings indicate that the rec-monkey FSHβ/α produced in this study demonstrated full bioactivity in stimulating cAMP production in cells expressing hFSHR. Moreover, cAMP responsiveness was significantly enhanced in the absence of β-arrestin 1, suggesting a regulatory role for β-arrestin 1 in FSHR-mediated signaling. This study highlights the potent biological activity of rec-monkey FSHβ/α and its role in promoting PKA-dependent signal transduction.

### 3.7. pERK1/2 Activation Through rLH/CGR, hLH/CGR, rFSHR, and hFSHR

To examine whether recombinant monkey LHβ/α and FSHβ/α induced ERK1/2 phosphorylation, pERK1/2 activation was first assessed in CHO-K1 cells expressing either human or rat gonadotropin receptors (hLH/CGR, rLH/CGR, hFSHR, or rFSHR) (Figure 7A,B). Treatment with concentrated (10-fold) recombinant monkey gonadotropins produced in CHO-K1 cells led to rapid pERK1/2 activation at 5 min in both hLH/CGR- and rLH/CGR-expressing cells, and this activation was sustained for up to 15 min. Thereafter, pERK1/2 levels declined to nearly baseline in hLH/CGR-expressing cells, whereas approximately 25% of the activity was retained in rLH/CGR-expressing cells. A similar pattern was observed in hFSHR- and rFSHR-expressing cells, although the magnitude of activation was lower than that in LH/CGR-expressing cells. At 30 min, pERK1/2 activity was still detectable at approximately 35–64% depending on the receptor subtype.

Next, pERK1/2 signaling was examined using purified recombinant monkey FSHβ/α produced in CHO-S cells and isolated via dual-affinity chromatography. HEK 293 cells transiently expressing either hFSHR or rFSHR were stimulated with 5 or 10 μL purified ligand. In hFSHR-expressing cells, rec-monkey FSHβ/α induced a rapid and robust increase in pERK1/2 phosphorylation that peaked at 5 min, followed by a sharp decline, indicating rapid desensitization of the signaling pathway (Figure 8A). In contrast, rFSHR-expressing cells exhibited a slower but more sustained activation pattern, with pERK1/2 levels peaking at 15 min and remaining at approximately 30–40% of the maximum value at 30 min post-stimulation (Figure 8B).

### 3.8. Inhibition of PKA and PKC Reveals Distinct Patterns of Rec-Monkey FSH-Induced pERK1/2 Activation

To further investigate the intracellular signaling mechanisms downstream of FSHR activation, we examined the contribution of two Gαs-associated pathways to ERK1/2 phosphorylation in response to recombinant monkey FSHβ/α. HEK293 cells expressing hFSHR were pretreated with the PKA inhibitor H-89 and stimulated with recombinant monkey FSHβ/α. Western blot analysis demonstrated that pERK1/2 activation at 5 min was strongly suppressed in the presence of H-89, indicating that early ERK1/2 signaling largely depends on PKA activity (Figure 9A). Although a weak residual signal was detectable at 15 min, its magnitude was substantially reduced relative to that in the untreated controls. Thus, FSH-induced ERK1/2 activation via hFSHR is predominantly mediated through the canonical cAMP/PKA pathway.

To test whether PKC contributes to ERK1/2 signaling, the cells were pretreated with the PKC inhibitor GFX before stimulation with the ligand. In contrast to H-89, GFX had little effect on the early pERK1/2 response at 5 min, and only a slight enhancement of phosphorylation was observed at 15–30 min (Figure 9B). These results indicate that PKC activity is not required for FSHβ/α-induced ERK1/2 activation in hFSHR-expressing cells and that the observed ERK1/2 signaling is largely insensitive to PKC inhibition.

### 3.9. Inhibition of pERK1/2 Activation in the β-Arrestin 1 Knockout Cells

To determine whether β-arrestin 1 contributes to ERK1/2 activation downstream of FSHR, we examined pERK1/2 phosphorylation in β-arrestin 1 knockout (KO) cells following stimulation with rec-monkey FSHβ/α. CRISPR-Cas9-mediated knockout of β-arrestin 1 was confirmed by Western blotting (Figure 10A). In control HEK293 cells expressing rFSHR, pERK1/2 levels increased robustly at 15 min post-stimulation and then declined by approximately 15% at 30 min, consistent with the sustained activation phase characteristic of rat FSHR signaling. This temporal pattern differed from that observed in hFSHR-expressing cells, where pERK1/2 levels peaked earlier at 5 min.

β-arrestin 1 KO cells exhibited markedly impaired ERK1/2 activation. One KO clone displayed an almost complete loss of pERK1/2 signaling across all time points, with only a minimal transient increase at 5 min, whereas the second KO clone reached only ~25% of the control response at 15 min (Figure 10B). These findings indicate that the sustained phase of ERK1/2 activation observed in rFSHR-expressing cells is β-arrestin-1-dependent.

Taken together, these results show that depletion of β-arrestin 1 abolished or severely reduced pERK1/2 activation, demonstrating that β-arrestin 1 is essential for FSH–FSHR-mediated MAPK pathway signaling and plays a critical role in sustaining ERK1/2 activation following FSH stimulation. These results support a signaling bias model in which FSHR uses both PKA- and β-arrestin-mediated pathways, with β-arrestin 1 specifically required for the sustained MAPK response.

## 4. Discussion

In this study, we re-characterized the nucleotide sequences of monkey gonadotropin cDNAs and compared them with previously reported data [22]. Three amino acid positions (44, 77, and 85) in the common α-subunit have been previously described as sites of interspecies variation between Cynomolgus and Rhesus monkeys; however, our sequencing results revealed no differences at either the nucleotide or amino acid level. These findings demonstrate that the common α-subunit is highly conserved between the two species, indicating strong evolutionary stability within the *Macaca* genus.

Comparative analysis revealed approximately 14 amino acid differences between the mature LHβ-subunits of monkey and human origin, suggesting moderate divergence that may contribute to species-specific receptor-binding characteristics or biological activity. Of the three subunits analyzed, the FSHβ subunit exhibited the highest degree of conservation between monkey and human sequences. Taken together, the evolutionary pressure to maintain functional integrity is strongest for the FSHβ-subunit, possibly reflecting the essential and conserved role of FSH in reproductive regulation across primate species.

The commercial ELISA kits used in this study failed to detect recombinant LHβ/α or FSHβ/α. The conformational epitopes recognized by ELISA antibodies may not be preserved or may be sterically masked in fusion proteins. Taken together, these results suggest that recombinant β/α single-chain fusion gonadotropins may exhibit distinct structural or post-translational characteristics compared to native dimeric hormones, which should be considered when selecting detection methods and designing functional assays.

The purified recombinant protein exhibited a broad molecular weight distribution, likely reflecting the heterogeneous glycosylation in CHO-S cells. This observation is consistent with the presence of N-linked oligosaccharide chains, which contribute to the increased molecular mass detected in the untreated samples. Previous studies have shown that the molecular weights of mammalian cell-derived glycoprotein hormones, including hFSH, hLH, hCG, eelLH, eelFSH, and eCG, are markedly reduced following PNGase F treatment [24,25,26,27]. Natural eCG reportedly contains O-acetylated sialic acids, whereas such modifications are not prominent in hCG, indicating that sialylation patterns can vary in a species-specific manner [28]. Consistent with this, we did not detect significant terminal sialic acid attachment in the rec-monkey FSHβ/α expressed in CHO-S cells. These findings suggest that the rec-protein produced in this study primarily carries N-linked rather than O-linked glycans. From a glycobiological perspective, the selective manipulation of O-linked glycosylation may provide a strategy for engineering gonadotropin variants with altered receptor interaction kinetics or prolonged biological activity.

In this study, we observed that cAMP responses were significantly higher in β-arrestin 1 KO cells than in WT cells. A similar effect has been reported for the 5A-FSHR mutant, in which five C-terminal phosphorylation sites of the rat FSHR were substituted with alanine [15]. The 5A-FSHR mutant displayed ~60% reduced agonist-induced receptor phosphorylation and an ~80% decrease in β-arrestin recruitment after 15 min of stimulation, resulting in a markedly increased maximal cAMP accumulation relative to the WT receptor. Likewise, mouse embryonic fibroblasts (MEFs) derived from β-arrestin 1/2 double knockout mice exhibited significantly elevated total cAMP accumulation following isoproterenol stimulation compared with WT MEF cells expressing either β_2_AR or AT_1_A-R [29]. Together, these findings demonstrate that the absence or reduction in β-arrestin recruitment enhances Gαs-mediated cAMP signaling, consistent with our observations in β-arrestin 1 KO cells. In contrast, the β_2_AR^TYY^ mutant (Thr68Ala, Tyr132Ala, and Tyr219Ala), which disrupts phosphorylation sites conserved across β_2_AR and rhodopsin, failed to induce cAMP accumulation, confirming the loss of Gαs-coupling capability [30]. Taken together, these observations suggest that mutations or loss of phosphorylation sites within the C-terminal tail of GPCRs can alter receptor function by modulating post-translational folding and β-arrestin recruitment. As β-arrestin 1 and β-arrestin 2 selectively bind phosphorylated GPCRs [8], the absence of β-arrestin 1 is expected to reduce phosphorylation-dependent receptor desensitization. Consistent with this interpretation, our results indicate that the deletion of β-arrestin 1 enhances cAMP signaling by preventing β-arrestin-mediated uncoupling of FSHR from Gαs. Thus, β-arrestin 1 functions as a negative regulator of FSHR-mediated cAMP production by promoting receptor desensitization.

Regarding pERK1/2 activity, our results indicated that rFSHR mediates a delayed but more sustained ERK1/2 activation response to recombinant monkey FSHβ/α than the human receptor (Figure 8A and Figure 9B). These findings suggest species-specific differences in receptor signaling dynamics, which may stem from variations in receptor conformation, intracellular regulatory machinery, or ligand–receptor interaction kinetics. Our observations align with previous reports showing that pERK1/2 activation peaks at approximately 10 min in rFSHR-expressing cells following agonist stimulation [15,31], whereas hFSHR—and most GPCRs—exhibit maximal pERK1/2 activation at around 5 min [31,32,33,34]. Therefore, to achieve a more comprehensive understanding of the FSH–FSHR signaling system, future studies should examine the temporal profile of ERK1/2 activation and the downstream pathways and regulatory mechanisms that contribute to sustained signaling.

In our experiments, treatment with H-89 markedly reduced pERK1/2 activation, whereas inhibition with GFX had little effect, indicating that ERK1/2 phosphorylation in this system is predominantly PKA-dependent rather than PKC-dependent. This result is consistent with previous studies demonstrating that the H-89-sensitive PKA-dependent G-protein-mediated signaling pathway regulates ERK1/2 activation downstream of β_2_AR [30], PTHR [33], and rat FSHR [15]. Accordingly, our findings suggest that rec-monkey FSHβ/α activates ERK1/2 via the Gαs-cAMP-PKA cascade in hFSHR-expressing HEK293 cells.

In this study, β-arrestin 1 KO HEK293 cells expressing rFSHR exhibited a dramatic reduction in pERK1/2 activation. These results are consistent with previous observations in rFSHR, β_2_AR, and PTHR models, in which β-arrestin 1 knockdown using siRNA led to attenuated ERK1/2 activation [15,30,33,35], reinforcing the critical role of β-arrestin 1 in mediating MAPK signaling downstream of FSHR. However, the effect of β-arrestin 1 depletion varies among GPCRs. Whereas siRNA-mediated β-arrestin 1 knockdown consistently reduced pERK1/2 activation in the β_2_AR, β_1_AR, V2R, and FSHR systems, β-arrestin 1 KO cells displayed divergent outcomes: increased pERK1/2 activation in β_2_AR, minimal change in β_1_AR and V2R, and markedly decreased activation in FSHR-expressing cells. These differences suggest that β-arrestin 1 contributes to ERK1/2 signaling through receptor-specific mechanisms and that ERK activation can proceed through distinct signaling pathways depending on the GPCR context [36].

For example, β_2_AR signaling to ERK has been shown to proceed independently of β-arrestin recruitment, instead using a Gαs/Gβγ–SRC–SHC–MAPK axis, whereas β-arrestin 2 remains essential for β_2_AR internalization [37]. In addition, increasing evidence demonstrates that β-arrestin-dependent signaling is highly receptor-specific and produces unique cellular outcomes, depending on receptor phosphorylation patterns and β-arrestin isoform engagement [12]. Furthermore, variations in GRK involvement contribute to this signaling diversity. GRK isoforms differentially regulate the formation and configuration of GPCR–β-arrestin complexes [38], and GRK2 plays a more prominent role than GRK3 in modulating downstream signaling outcomes [39].

Based on our results, rec-monkey FSHβ/α stimulated dose-dependent cAMP accumulation in HEK293 cells expressing hFSHR, demonstrating that the recombinant protein retained its full biological activity. These data establish a reliable system for producing biologically active recombinant monkey gonadotropins and demonstrate that β-arrestin 1 exerts a dual regulatory role by suppressing Gαs-driven cAMP accumulation while being essential for sustained ERK1/2 activation. This study advances our understanding of the structure–function relationships of primate gonadotropin and provides new insights into the mechanism by which β-arrestin 1 modulates the signaling bias at FSHR. β-arrestin 1 functions as a key determinant of signaling bias at FSHR, suppressing Gαs-mediated cAMP signaling while enabling sustained ERK1/2 activation.

## 5. Conclusions

We successfully generated biologically active recombinant monkey FSHβ/α and defined its signaling characteristics in FSHR-expressing cells. Notably, rFSHR exhibited a delayed yet more sustained ERK1/2 response compared to hFSHR, highlighting species-specific differences in receptor activation dynamics. Pharmacological inhibition demonstrated that FSH-driven ERK1/2 activation is governed predominantly by the cAMP/PKA axis, with minimal contribution from PKC. Mechanistically, β-arrestin 1 emerged as a critical bifunctional regulator: its deletion enhanced Gαs-mediated cAMP production while completely abolishing ERK1/2 activation. These results collectively reveal that β-arrestin 1 orchestrates a dual regulatory network within the FSHR signaling cascade and that FSH–FSHR signaling integrates both classical G-protein-dependent and β-arrestin-dependent pathways. To produce recombinant monkey FSH with higher biological activity, it is essential to establish a production system that enables proper O-linked glycosylation. Our findings not only elucidate key mechanisms underlying primate FSHR signaling but also provide a robust foundation for future therapeutic and biotechnological applications involving recombinant gonadotropins.

## Figures and Tables

**Figure 1 cimb-47-01051-f001:**
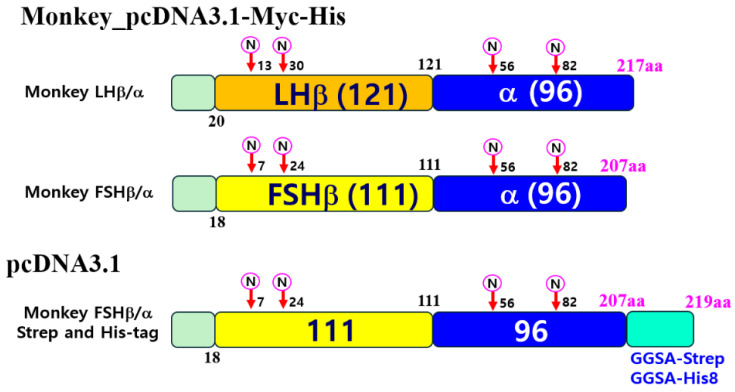
Schematic representation of recombinant monkey LHβ/α, FSHβ/α, and FSHβ/α constructs with Strep- or His-tag. Single-chain forms of LHβ/α and FSHβ/α were generated by overlapping PCR, as described in the Materials and Methods. The C-terminal region of each β-subunit was directly fused to the N-terminus of the α-subunit lacking its native signal peptide. The resulting β/α single-chain genes were first subcloned into the pcDNA3.1(-)/Myc-His mammalian expression vector. In addition, the FSHβ/α gene was subcloned into pcDNA3.1(-) vectors containing either a GGSA-Strep-tag or a His-tag at the C-terminus of the α-subunit. A circled “N” indicates predicted N-linked glycosylation sites. The final expression constructs were designated as pcDNA3.1(-)/Myc-His-M-LHβ/α, pcDNA3.1(-)/Myc-His-M-FSHβ/α, and pcDNA3.1-M-FSHβ/α-Strep, pcDNA3.1-M-FSHβ/α-His. LHβ: Orange; α: Blue; FSHβ: Yellow; Strep and His-tags: Cyan; Signal Sequence: Mint.

**Figure 2 cimb-47-01051-f002:**
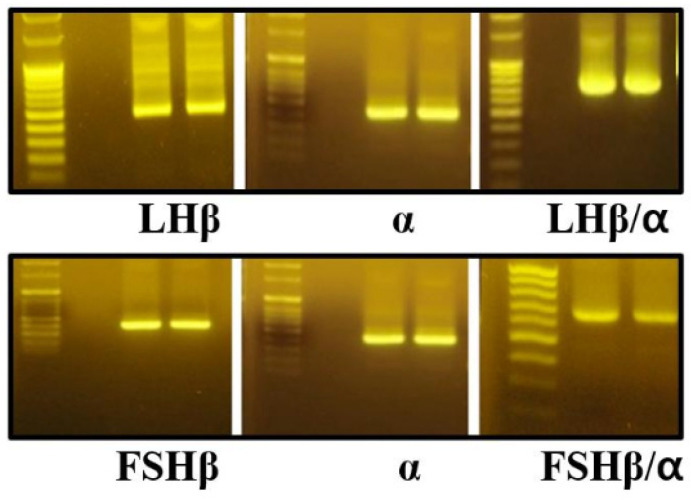
PCR amplification of monkey pituitary cDNA samples. Total RNA was isolated from monkey pituitary tissue and reverse-transcribed to generate first-strand cDNA. Primers containing restriction enzyme recognition sites for EcoRI and SalI or XhoI were designed at the 5′ and 3′ ends of each gene. The first-round PCR was performed under optimized conditions to specifically amplify the individual subunit cDNA fragments. The amplified sizes of the α-, LHβ-, and FSHβ-subunits were 375 bp, 437 bp, and 404 bp, respectively. After sequence verification, a second-round PCR was carried out to generate the single-chain LHβ/α and FSHβ/α constructs.

**Figure 3 cimb-47-01051-f003:**
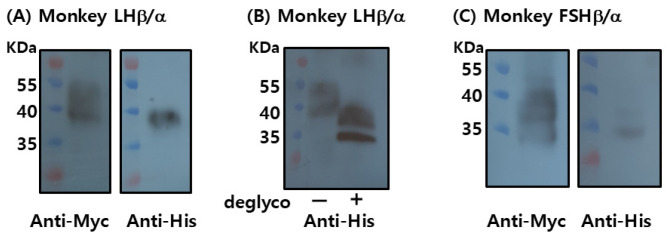
Western blot analysis of recombinant monkey LHβ/α and FSHβ/α proteins. Recombinant proteins were expressed in CHO-K1 cells. (**A**) LHβ/α proteins were separated by SDS-PAGE and detected using monoclonal anti-Myc and anti-His antibodies. (**B**) LHβ/α proteins were treated with PNGase F for 1 h at 37 °C, resulting in a mobility shift indicative of N-linked glycan removal. (**C**) FSHβ/α proteins were similarly analyzed by SDS-PAGE and detected using monoclonal anti-Myc and anti-His antibodies. “−” indicates untreated samples, and “+” indicates PNGase F-treated samples.

**Figure 4 cimb-47-01051-f004:**
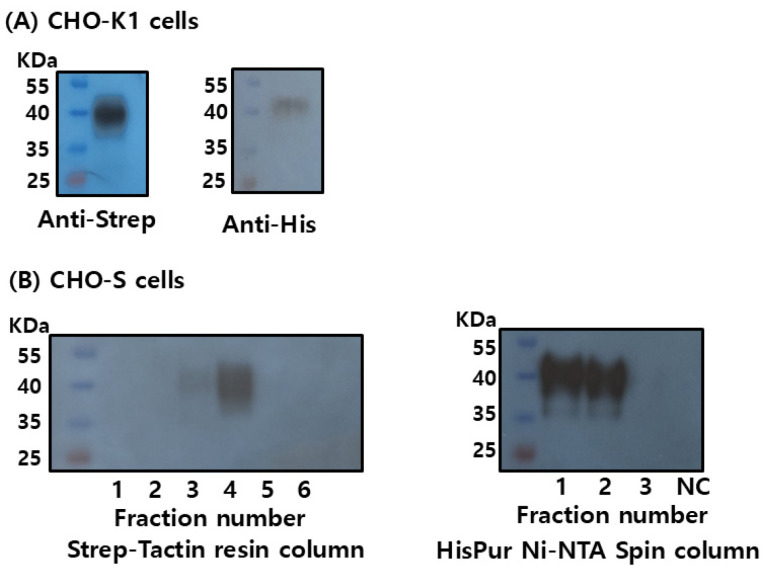
Western blot analysis and purification of recombinant monkey FSHβ/α using dual epitope tagging in CHO-K1 and CHO suspension cells. The single-chain FSHβ/α cDNA was subcloned into the pcDNA3.1 mammalian expression vector and transfected into CHO-K1 adherent cells and CHO suspension cells. Culture supernatant from CHO-K1 cells was concentrated approximately 10-fold prior to analysis. Recombinant FSHβ/α proteins were detected using anti-Strep and anti-His monoclonal antibodies (**A**). Recombinant FSHβ/α secreted from CHO suspension cells was purified using Strep-Tactin affinity resin ((**B**), **left**) and HisPur™ Ni-NTA spin columns ((**B**), **right**).

**Figure 5 cimb-47-01051-f005:**
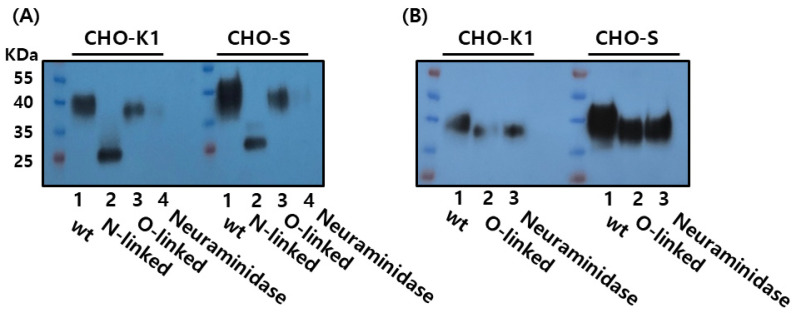
Enzymatic deglycosylation of recombinant monkey FSHβ/α by PNGase F, Endo-α-N-acetylgalactosaminidase, and neuraminidase. Recombinant FSHβ/α proteins expressed in CHO-K1 and CHO-S cells were treated with PNGase F, resulting in a marked reduction in molecular weight to approximately 25–26 kDa, indicating the presence of N-linked glycans (**A**). In contrast, treatment with Endo-α-N-acetylgalactosaminidase (O-glycosidase) or neuraminidase did not produce a noticeable shift in molecular weight, suggesting minimal O-linked glycosylation on the FSHβ/α protein (**B**).

**Figure 6 cimb-47-01051-f006:**
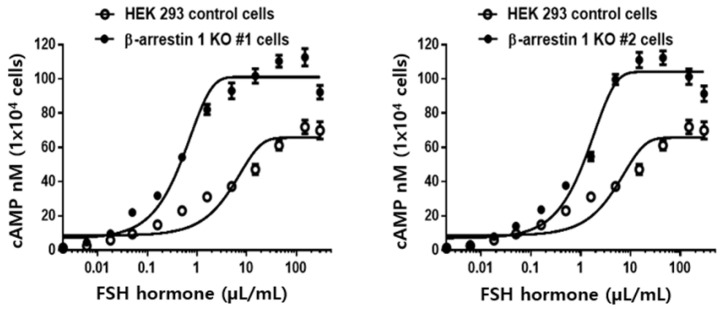
Dose-dependent increase in total cAMP accumulation induced by recombinant monkey FSHβ/α in wild-type and β-arrestin 1 knockout cells expressing the human FSH receptor. Cells stably expressing the hFSH receptor were seeded at 10,000 cells per well in 384-well plates. Serial dilutions of recombinant FSHβ/α (0.17–712 nM) were applied to cells and incubated for 30 min at room temperature in the presence of 0.5 mM IBMX. Following stimulation, cAMP-d2 and cAMP Eu-Cryptate antibodies were added, and fluorescence signals were measured at 665 nm and 620 nm using an HTRF microplate reader. cAMP levels were quantified using GraphPad Prism, based on standard curve fitting. Data are presented as total cAMP accumulation per 1 × 10^4^ cells. Data are presented as the mean ± SEM from three experiments. KO #1 and #2 denote two independently isolated cell lines.

**Figure 7 cimb-47-01051-f007:**
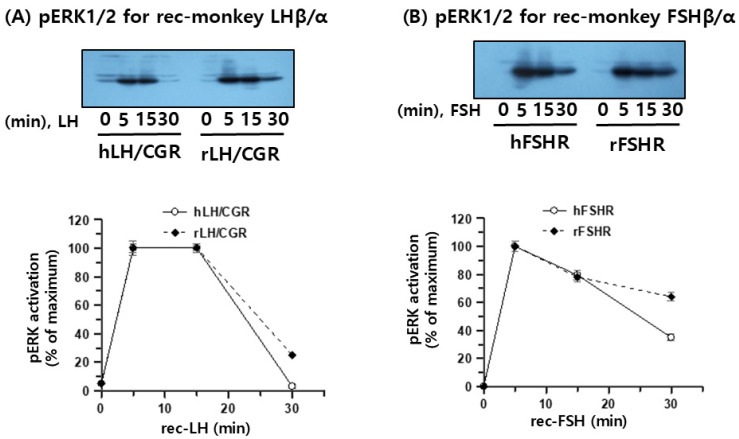
pERK1/2 activation mediated by hLH/CGR, rLH/CGR, hFSHR, and rFSHR. HEK293 cells were transiently transfected with each receptor construct, serum-starved for at least 6 h, and then stimulated with recombinant monkey LHβ/α or FSHβ/α produced in CHO-K1 cells. Cell lysates were prepared following stimulation, and 10 µg of total protein per sample was subjected to SDS-PAGE and immunoblotting for phosphorylated ERK1/2. Representative blots are shown for the time-dependent activation of ERK1/2 in cells expressing hLH/CGR and rLH/CGR (**A**) and hFSHR and rFSHR (**B**). pERK1/2 band intensities were quantified by densitometry, and the maximal phospho-ERK1/2 signal (typically at 5 min) was normalized to 100%. Data are presented as the mean ± SEM from three independent experiments.

**Figure 8 cimb-47-01051-f008:**
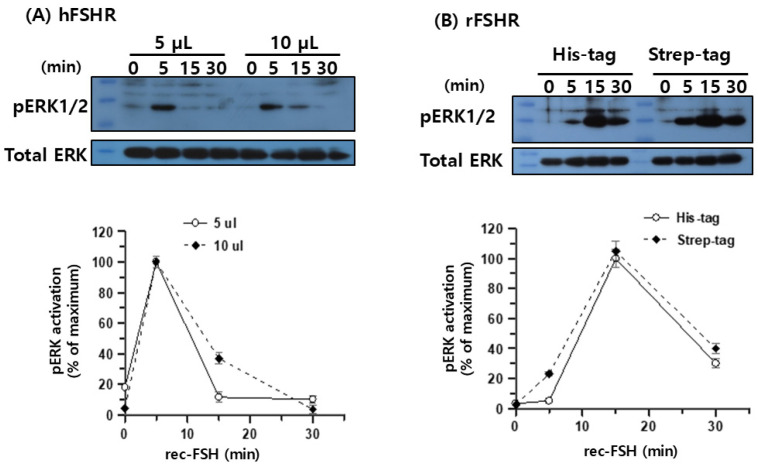
pERK1/2 activation induced by recombinant monkey FSHβ/α in cells expressing human and rat FSH receptors. HEK293 cells were transiently transfected with hFSHR or rFSHR and serum-starved for 6 h prior to stimulation. Cells were then stimulated with 5 µL or 10 µL of purified recombinant monkey FSHβ/α at 37 °C for the indicated times. pERK1/2 activation was assessed by immunoblot analysis, and band intensities of pERK1/2 were normalized to total ERK. For cells expressing hFSHR, the maximal pERK1/2 response occurred at 5 min, and this value was set to 100% (**A**). In contrast, in cells expressing rFSHR, the maximal response was observed at 15 min, which was normalized to 100% (**B**). Data represent the mean ± SEM from three independent experiments.

**Figure 9 cimb-47-01051-f009:**
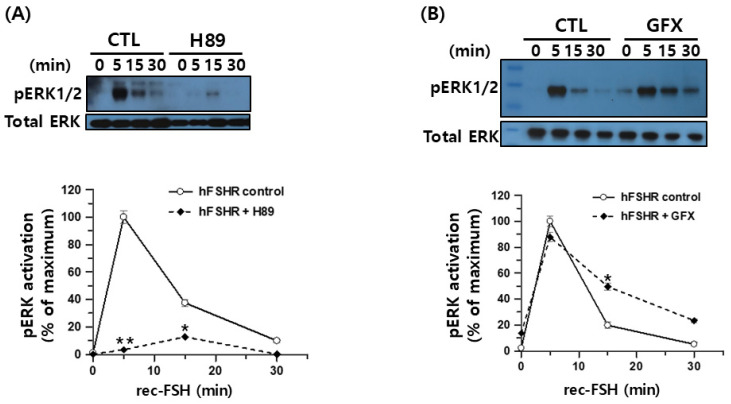
Sensitivity of hFSHR-mediated pERK1/2 activation to the PKA inhibitor H-89 and the PKC inhibitor GFX in HEK293 cells. HEK293 cells were transiently transfected with hFSHR and serum-starved for 6 h prior to stimulation. Cells were pretreated with 20 µM H-89 for 15 min (**A**) or 2.5 µM GFX for 15 min (**B**), followed by stimulation with recombinant monkey FSHβ/α. pERK1/2 levels were measured by immunoblotting, and band intensities were normalized to total ERK. The maximal pERK1/2 response at 5 min in the absence of inhibitor was set to 100%. Data represent the mean ± SEM from three independent experiments. Statistical significance between inhibitor-treated and untreated groups over the time course was assessed using two-way ANOVA. * *p* < 0.05; ** *p* < 0.01 vs. control curve.

**Figure 10 cimb-47-01051-f010:**
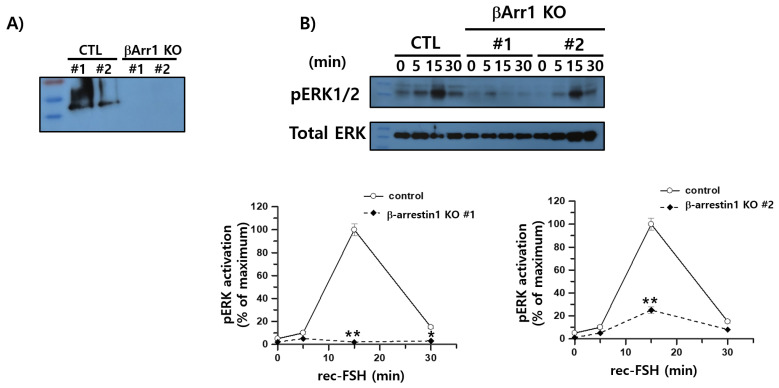
Effects of β-arrestin 1 knockout on ERK1/2 activation mediated by the rFSHR. (**A**) Successful deletion of β-arrestin 1 in CRISPR-edited cells was confirmed by Western blotting using an anti-β-arrestin 1 antibody. (**B**) HEK293 control cells and β-arrestin 1 knockout cells were transiently transfected with rFSHR and serum-starved for 6 h prior to stimulation with recombinant monkey FSHβ/α for the indicated times. Cell lysates were analyzed by immunoblotting for pERK1/2 and total ERK. pERK1/2 signals were normalized to total ERK, and the maximal pERK1/2 response observed in control cells was set to 100%. Data are presented as mean ± SEM from three independent experiments. Statistical significance between control and β-arrestin 1 knockout groups was determined using two-way ANOVA (*n* = 3). * *p* < 0.05; ** *p* < 0.01 vs. control curve. KO #1 and #2 denote two independently isolated cell lines.

**Table 1 cimb-47-01051-t001:** Bioactivity of monkey rec-FSH in cells expressing hFSH receptor.

Cell Lines	cAMP Responses		
	Basal *^a^* (nM/10^4^ Cells)	EC_50_ (µL/mL)	Rmax *^b^* (nM/10^4^ Cells)
HEK 293	8.6 ± 1.6	4.9 (1.0-fold) (3.5 to 8.0) *^c^*	66.7 ± 2.4 (1.0-fold)
β-arrestin 1 KO 1	6.8 ± 1.6	0.5 (9.8-fold) (0.4 to 0.6)	101.9 ± 1.7 (1.52-fold)
β-arrestin 1 KO 2	6.9 ± 1.7	1.2 (4.1-fold) (1.1 to 1.6)	104.2 ± 1.9 (1.56-fold)

Values are the means ± SEM of three experiments. The half-maximal effective concentration (EC_50_) values were determined from the concentration-response curves from in vitro bioassays. *^a^* Basal cAMP level average without agonist treatment. *^b^* Rmax average cAMP level/10^4^ cells. *^c^* Geometric mean (95% confidence limit).

## Data Availability

The original contributions presented in this study are included in the article/Appendix A. Further inquiries can be directed to the corresponding author.

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
