# Peer review of "β-Arrestin 1 Differentially Modulates cAMP and ERK Pathways Downstream of the FSH Receptor"

_cimb, 2025, doi:10.3390/cimb47121051_

Round 1
Reviewer 1 Report
Comments and Suggestions for Authors
-Scientifically valid, but major revision for clarity, conciseness, and strengthened data interpretation
-Non-specific comments
- Cross-check the references in the text and the full reference
- Cite the figures and tables in the results and discussion.
- check Gene's name style
-Abstract
-Too long and dense abstracts should be more concise.
-Methodological details, such as CHO cell lines, deglycosylation, which are not needed in the abstract
-Strongly descriptive but lacks a clear statement on significance or novelty
-Keywords
Keywords should be more specific to the title. I suggest picking a more suitable one
-Introduction
- Duplicate explanations of β-arrestin scaffolding and internalization roles appear multiple times.
- The rationale for needing monkey recombinant hormones is not emphasized enough until late in the introduction.
Materials and methods
-Some figures mention three replicates, others do not
-Why are specific phosphorylation inhibitors (H-89 and GFX)
-Two-way ANOVA is appropriate, but more clarity is needed regarding sample size (n)
-Provide a clearer workflow diagram for recombinant hormone production
- Primer validation is needed, and the efficiency and the internal control efficiency are needed.
-Discussion
-Too long and repetitive, recaps result in detail instead of critically analyzing them
-Discussion heavily focuses on these two aspects, i.e., ERK and cAMP, while lacking the rest of the phenomena.
-Evaluate in primary gonadal cells
-test downstream gene expression responses, e.g., aromatase, inhibin
Conclusion
-Mostly repeats results without highlighting broader impact
-Mention applicability for developing improved FSH analogs
Author Response
Comments and Suggestions for Authors
-Scientifically valid, but major revision for clarity, conciseness, and strengthened data interpretation
-Non-specific comments
- Cross-check the references in the text and the full reference
- Cite the figures and tables in the results and discussion.
- check Gene's name style
-Abstract
-Too long and dense abstracts should be more concise.
-Methodological details, such as CHO cell lines, deglycosylation, which are not needed in the abstract
-Strongly descriptive but lacks a clear statement on significance or novelty
→ I have summarized the abstract in accordance with the reviewer’s comments.
-Keywords
Keywords should be more specific to the title. I suggest picking a more suitable one
→ We change the keywords by reviewer’s comments.
-Introduction
- Duplicate explanations of β-arrestin scaffolding and internalization roles appear multiple times.
→We also deleted “β-arrestins function as multifunctional scaffolding proteins that not only desensitize and terminate G-protein signaling but also initiate distinct downstream signaling cascades, FSHR interacts with β-arrestins, which serve as scaffolding proteins that regulate receptor desensitization, internalization, and recycling.” In the Introduction section.
- The rationale for needing monkey recombinant hormones is not emphasized enough until late in the introduction.
→We inserted “Due to the recent increase in new drug development, the demand for monkeys has risen sharply, leading to a growing need for monkey FSH for controlled ovarian stimulation. Therefore, it is necessary to develop a large-scale production system for monkey FSH to meet this demand.” In the introduction section Line 84-87.
Materials and methods
-Some figures mention three replicates, others do not
→All data obtained from three replicate experiments. Thus, we inserted the “Data are presented as the mean ± SEM from three experiments.” In Figure 6 Legend.
-Why are specific phosphorylation inhibitors (H-89 and GFX)
→ FSH binding to FSHR activates both the PKA and PKC signaling pathways. To determine which pathway has a greater impact on pERK1/2 activation, we treated cells with specific inhibitors. The results showed that inhibition of PKA suppressed FSHR-mediated pERK1/2 activation is more strongly than inhibition of PKC.
-Two-way ANOVA is appropriate, but more clarity is needed regarding sample size (n)
→We inserted (N= 3) in Figure 10 Legend.
-Provide a clearer workflow diagram for recombinant hormone production
→ As the detailed procedures were presented in our previous report (Ref. 23), the methods in this paper are described more concisely for brevity
- Primer validation is needed, and the efficiency and the internal control efficiency are needed.
→ The primer sequences are provided in Supplementary Tables 1 and 2. However, we believe that further details, such as internal control efficiency, are not required for the purposes of this study.
-Discussion
-Too long and repetitive, recaps result in detail instead of critically analyzing them
→Deleted the repeated regions.
-Discussion heavily focuses on these two aspects, i.e., ERK and cAMP, while lacking the rest of the phenomena.
→We are revised in many respects.
-Evaluate in primary gonadal cells
→We didn’t any experiments for primary gonadal cells, thus in this study we focus on FSHR-mediated signaling in vitro.
-test downstream gene expression responses, e.g., aromatase, inhibin
→In the presented study, we also didn’t have any gene expression related to downstream signaling.
Conclusion
-Mostly repeats results without highlighting broader impact
→The conclusion section was rewritten.
-Mention applicability for developing improved FSH analogs
→We inserted “To produce recombinant monkey FSH with higher biological activity, it is essential to establish a production system that enables proper O-linked glycosylation.” In the conclusion section.

Reviewer 2 Report
Comments and Suggestions for Authors
Dear Authors,
The article “β-Arrestin 1 Differentially Modulates cAMP and ERK Pathways Downstream of the FSH Receptor» focuses on robust system for the production of biologically active recombinant monkey gonadotropins. The authors revealed dual roles of β-arrestin 1 in modulating FSHR-mediated cAMP and ERK signaling pathways
All sections of the manuscript are thoroughly and clearly written. I have no substantive criticisms to note. Statistical analyses were performed satisfactory.
The discussion and introduction would benefit from the inclusion of the following aspects to enhance their rigor and impact.
- For the Introduction section (background).
From a practical perspective, which signaling pathway of β-Arrestin 1 holds greater importance for pharmaceutical development?
- For the Methods section.
What was the reason to compare the sequence differences between cynomolgus and Rhesus monkeys, please, explain the choice of these spices.
- For the Discussion section.
It is known that arrestin initiates alternative signaling pathways. Could they have influenced your results?
- For the Discussion section.
From a drug development perspective, what conclusion can be drawn from the data obtained? That is, is there any practically significant result for pharmacology?
Author Response
Comments and Suggestions for Authors
Dear Authors,
The article “β-Arrestin 1 Differentially Modulates cAMP and ERK Pathways Downstream of the FSH Receptor» focuses on robust system for the production of biologically active recombinant monkey gonadotropins. The authors revealed dual roles of β-arrestin 1 in modulating FSHR-mediated cAMP and ERK signaling pathways
All sections of the manuscript are thoroughly and clearly written. I have no substantive criticisms to note. Statistical analyses were performed satisfactory.
The discussion and introduction would benefit from the inclusion of the following aspects to enhance their rigor and impact.
- For the Introduction section (background).
From a practical perspective, which signaling pathway of β-Arrestin 1 holds greater importance for pharmaceutical development?
→ Of course, cAMP signaling is the best way to analyze the biological activity of rec-glycoprotein hormones. β-Arrestin 1 signaling is also very important for MAPK signaling pathway. Thus, we explained the point of β-Arrestin function in an intermediate parts abstract section.
- For the Methods section.
What was the reason to compare the sequence differences between cynomolgus and Rhesus monkeys, please, explain the choice of these spices.
→OK, we inserted the reason why monkey glycoprotein hormone genes choice.
We inserted “Previous studies have indicated that the glycoprotein hormone sequences differ between cynomolgus and rhesus monkeys. To confirm these observations, we conducted an additional comparative sequence analysis of the two species.” In the Methods section.
- For the Discussion section.
It is known that arrestin initiates alternative signaling pathways. Could they have influenced your results?
→We also explained the functions of β-Arrestin 1 KO cells in the Discussion section. However, the cAMP level for PKA signaling pathway in β-Arrestin 1 KO cells increased inversely. These results are accepted as new phenomenon.
- For the Discussion section.
From a drug development perspective, what conclusion can be drawn from the data obtained? That is, is there any practically significant result for pharmacology?
→From a drug development perspective, rec-monkey FSH are showing the physiological activity for cAMP and pERK1/2 through FSHR. Now, we are going to produce rec-monkey FSH in the CHO DG44 cells.
→Thus, we explained “the system for producing biologically active recombinant monkey gonadotropins.

Round 2
Reviewer 1 Report
Comments and Suggestions for Authors
The effort is commendable, and I accept the manuscript